Multipath subflow transmission scheduling optimization algorithm based on cost-performance balance

Sun Xinyu sxy007@jcut.edu.cn
College of New Energy, Jingchu University of Technology , Jingmen , China
Sohaib Osama
Electronic publication date: 2025 Apr 24
Publication date: 2025
Volume: 11
Electronic Location ID: e2838
Received 2025 Jan 25; Accepted 2025 Mar 26
Copyright: © 2025 Sun
Copyright year: 2025
Copyright holder: Sun
License: This is an open access article distributed under the terms of the Creative Commons Attribution License, which permits unrestricted use, distribution, reproduction and adaptation in any medium and for any purpose provided that it is properly attributed. For attribution, the original author(s), title, publication source (PeerJ Computer Science) and either DOI or URL of the article must be cited.
License URL: https://creativecommons.org/licenses/by/4.0/

Keywords: Multipath transmission control protocol, Software-defined, Networking, Generation V communication network, Subflow buffer queue, Cost-performance balance control

Funding: The authors received no funding for this work.

==============================
This article proposes a cost-performance balance algorithm for multipath data transmission within an Software Defined Network (SDN)-5G-Multipath Transmission Control Protocol (MPTCP) network framework. A unified communication interface is designed to schedule transmission paths, optimizing data flow allocation dynamically. To achieve a balance between network throughput and transmission costs, traffic volume and associated expenses are mapped into physical and virtual buffer queues for real-time substream updates. The relationship between unreceived subflows and consumption costs is mathematically modelled, with both parameters represented using vector matrices. Lyapunov stability theory is applied to determine the optimal cost-performance balance. In contrast, key evaluation metrics—including substream transmission efficiency, consumption expenditure, and overall balance control—are introduced to assess performance. Comparative analysis against classical price balance control algorithms demonstrates that the proposed strategy significantly enhances data transmission efficiency while reducing costs. Experimental results validate the effectiveness of the model in achieving cost-performance balance control for multi-channel transmission of large-scale files (ranging from 2 to 20 GB) under varying network conditions. The findings highlight the potential of this approach for optimizing high-performance, cost-efficient data transmission in next-generation communication networks.

Introduction

Under the Software-Defined Wireless Network (SDWN) platform, when using Multipath Transmission Control Protocol (MPTCP) (Sun et al., 2022; Wang, Zhang & Zhao, 2017; Li, Chen & Jiang, 2019; Jiang, Zhao & Li, 2019; Zhang et al., 2023), the MPTCP subflow scheduling allocation not only needs to consider the main factor of data transmission performance but also the consumption factor. The consumption of data transmission mainly includes two aspects: one is the consumption of hardware resources. Many network resources are enabled in the wireless network data transmission path, and many data transmission paths are provided. The utilization of network resources is high, and the network operation performance is good, but the power consumption is large. The other is service expenditure consumption. Currently, in the enjoyment of Internet data services, all kinds of data received are usually consumed, especially the large number of services provided in the current promotion of 5G communications (Qu, Wang & Zhou, 2015; Teng, Sun & Yang, 2020). Figure 1 shows the currently widely used 5G communication network structure and major service areas. Therefore, achieving a balance between data transmission performance and consumption control is also one of the main directions of SDWN technology application research.

Figure 1 SDN-5G-MPTCPM network data transmission process diagram.

In 5G communication networks, the key technology used is Software-Defined Network (SDN) technology, and the key point of technology application lies in network resources and data flow scheduling (Bruno Astuto, Mendes & Barroso, 2014; Bernardos, Zhao & Chen, 2014; Kai, Liu & Sun, 2017). The SDN technology uses the OpenFlow communication rules, which mainly control the connection between the SDN controller and the OpenFlow switch. It is necessary first to solve the problem of the connection between the SDN network and the 5G communication network terminal (Sun et al., 2024; Rodriguez-Natal, Gonzalez & Vega, 2018) and then realize the communication and data transmission of the SDWN network. Since the performance evaluation of data communication networks focuses on the reliability and stability of data transmission (Zhou, Zhao & Zhang, 2017; Zhou, Yang & Chen, 2017), MPTCP communication strategies have been widely used to achieve optimal scheduling between transmission data and transmission paths (Paasch, Zhang & Wang, 2014). There are many classical data transmission optimization scheduling algorithms for MPTCP (Xue, Li & Liu, 2017; Frommgen, Heuschkel & Koldehofe, 2018; Matsufiiji, Tanaka & Lin, 2017; Kimura, Lima & Loureiro, 2017), which mainly aim to improve network throughput and reduce network transmission delay and almost do not involve the consumption of data transmission. We have previously carried out a detailed analysis of the optimization scheduling problem between MPTCP transmission data and transmission path and carried out relevant research; this article is not too much to introduce, focusing on the balance between data transmission performance and consumption control.

In recent years, with the rapid development and promotion of data centres and data services, the problem of consumption control in the operation of 5G communication networks has been paid attention to, and the corresponding research breakthrough points have been fully considered, fully considering the balance between hardware resources and service expenditure consumption. There is literature (Tang, Sun & Wu, 2019), for the virtual deployment in the 5G network, dividing physical resources, introducing priority for resource scheduling, and building an algorithm model for the two parameters of minimum power consumption and expenditure consumption achieve a balance between improving resource utilization and expenditure consumption. There is literature (Cao, 2020), given the problem of resource consumption in wireless networks, a standard network model is constructed according to the blockchain process, and the data transmission and data processing consumption of the two targets, proof of workload (PoW) and practical Byzantine fault tolerance (PBFT), are analyzed to verify the impact of their target consumption on network performance. There is literature (Hua, Zhang & Liu, 2020), given the lack of continuity of communication between the two sides of the wireless fusion network, the network organization structure and computer network construction theory of the biological physarum polycephalum are used to introduce an intelligent routing mechanism, build a wireless fusion network model, take the capacity of the data transmission path as the breakthrough, design the best route control strategy, and achieve efficient data transmission in the case of low network overhead. There is literature (Yuan & Huang, 2022) aiming at the route control strategy of establishing a balance between network performance and network redundancy in mobile ad hoc networks, the reasons for excessive power consumption and expenditure consumption are analyzed, and the balance between network transmission performance and network energy consumption is achieved by calculating and making full use of the remaining transmission capacity in the network.

This article focuses on the control of service expenditure consumption, makes full use of the advantages of MPTCP multipath data subflows transmission in 5G communication networks, designs a multi-channel transmission data transmission scheduling algorithm by constructing a network model of MPTCP and OpenFlow fusion, and introduces an MPTCP subflow transmission consumption control mechanism to alleviate the contradiction of the price balance of data transmission.

Network model

The network model designed in this article mainly includes five parts: MPTCP data subflow sender (MPTCP-S), MPTCP data subflow receiver (MPTCP-R), SDN controller (SDN-C), OpenFlow switch (OpenFlow) and 5G communication network architecture (5G-N). MPTCP-S completes the multi-channel transmission data transmission scheduling control, and the main task is to implement queue management for the transmission path of the MPTCP data subflow. MPTCP-R queues the MPTCP data subflow in an orderly manner according to the defined transmission path and pays the MPTCP sub-traffic transmission cost. SDN-C does not directly control MPTCP-S and MPTCP-R but is only responsible for controlling 5G-N, deploying OpenFlow switches in 5G-N, and providing multipath MPTCP subflow transmission channels that can treat SDN-C and 5G-N as one.

In the multi-transmission data transmission scheduling policy of the MPTCP data subflow sender, the application layer (App-L) transmission task is implemented MPTCP subflow encapsulation (MPTCP-E), multiple network connection interfaces (MPTCP-I) is provided, the network connection permission policy (MPTCP-Y) is imported. The MPTCP subflow allocation policy (MPTCP-P) is formulated, and the transmission path queue (MPTCP-Q) is constructed. In the transmission path queue, three types of buffers are designed: MPTCP sub-traffic cost data physical buffer (A-F ¥), MPTCP sub-traffic cost virtual buffer (a-F ¥), and MPTCP data subflow data buffer (D-MPTCP). The purpose of the MPTCP sub-traffic cost virtual buffer is to improve the storage needs of the MPTCP sub-traffic cost update changes and can also improve the stability of the entire transmission path queue. The MPTCP data feed data buffer consists of multiple physical buffers that provide multipath MPTCP subflow transmission. The above three types of buffers are used as a complete path queue for data transmission, and the MPTCP sub-traffic cost is paid by the receiving end of the MPTCP data subflow, and the cost paid includes part of the resource consumption cost.

In the entire network, the set of paths defining the transmission of all information is J, and J={1,2,…,j,…,J}; the set of interfaces defining the network connection is I, and I={1,2,…,i,…,I}, which is used to represent the number of different connection paths; the time of all information transmission is discretely defined, a certain moment is defined as t, the set of time is T, and t∈{1,2,…,T}. The data transmission process can be simply described by using the jth path of the ith interface at the t moment to transmit the MPTCP subflow. Define the number of MPTCP subflows transmitted using the ith interface at the t time is W(i,t); due to the use of network connection licensing policy, the actual number of encapsulated subflows W(i, t) includes a certain number of subflows that cannot be transmitted normally. Additionally, we define the number of subflows that are successfully transmitted as U(i, t); MPTCP subflow transmission consumption cost is related to the number of subflows, the number of transmission paths, and the consumption cost of occupying a physical buffer is denoted by B((A−F¥)(i,j,t)), where (A−F¥) represents the physical buffer consumption cost at time t for the subflow transmitted through the i-th interface and j-th path. According to the parameters defined above, the network model designed in this article is shown in Fig. 1, called the SDN-5G-MPTCPM network model.

Data transfer and consumption control policies

Abreviations and acronyms

The formulation of the network connection license policy MPTCP-Y needs to be defined according to the number of subflows received by the receiving MPTCP-R U(i,t) and the number of subflows encapsulated by the subflow sender MPTCP-S W(i,t), each communication can only allow the number of U(i,t) subflows into the buffer queue. Therefore, the key conditions of the network connection licensing policy are 0 < U(i,t) < W(i,t), and the transmission of all U(i,t) number of subflows needs to be completed, that is, the existence of W(i,t)=∑i=1,t=1I,TU(i,t).

Several important parameters are introduced in the proposed algorithm to model and control the balance between network transmission performance and consumption cost. These parameters include: α is an evaluation factor used in the throughput evaluation function. It plays a critical role in measuring network performance. β is a control parameter that balances network performance and transmission consumption cost. It defines the trade-off between throughput and cost, with a range from 0 to infinity. When β = 0, transmission costs are not considered, while β→∞ prioritizes transmission performance over cost. δ is an evaluation parameter for the normal transmission of subflows. It determines the level of normal reception for subflows in the transmission process. The range of δ is from 0 to 1, with 0 indicating poor reception performance and 1 indicating perfect reception. ε represents the error or variation between consecutive moments for subflow reception. It measures the change in subflow transmission performance over time and plays a role in assessing the stability and reliability of data transmission. The range for ε is between 0 and 1, with smaller values indicating more stable and reliable transmission. λ is the evaluation parameter for balancing network transmission throughput and consumption expenditure. It controls how much the algorithm focuses on improving throughput vs minimizing cost. The range of λ is from 0 to 1, where smaller values favour throughput, and larger values favour reducing costs.

In the scheduling of the data transmission path, the MPTCP subflow is transmitted on the j path through i interfaces, and at the t-time, according to the network connection permission conditions, the number of subflows transmitted on the j path through i interfaces is U(i,j,t), and a condition must be met W(i,j,t)=∑i=1,j=1,t=1I,J,T⁡U(i,j,t). At this point, the number of subflows in the buffer is W(i,j,t), and the number of subflows that MPTCP-R actually receives normally is U(i,j,t).

In the data transmission consumption cost payment, according to the definition of the data transmission path above, at the t time, the consumption cost of the physical buffer occupied by the MPTCP subflow transmission on the j path through i interface is B((A−F¥)(i,j,t)) and the total consumption cost is defined as B((A−F¥)(t)), then B((A−F¥)(t))=∑i=1,j=1,t=1I,J,TB((A−F¥)(i,j,t)). The consumption cost of occupying a virtual buffer is B((a−F¥)(i,j,t)), and the total consumption cost of the definition is B((a−F¥)(t))=∑i=1,j=1,t=1I,J,TB((a−F¥)(i,j,t)).

Transport update policy

The data transfer process also involves the real-time update of the MPTCP subflow queue in the physical buffer and the consumption cost queue update. At the t-time, the number of subflows in the buffer is W(i,j,t), and the update of the subflow is to calculate the number of subflows at the next moment t + 1 W(i,j,(t+1)), then there is:

(1) W(i,j,(t+1))=max[W(i,j,t)−U(i,j,t),W(0,0,0)]+U(i,j,t)

where: W(i,j,t) represents the number of subflows in the buffer at a time.

U(i,j,t) is the number of subflows successfully received.

The term W(0,0,0) represents the state where no data is being transferred.

It updates the number of subflows in the buffer based on the number of successfully received subflows. In Formula (1), max[W(i,j,t)−U(i,j,t),W(0,0,0)] the maximum value in the value range is represented, indicating the state of no data transfer W(0,0,0)=0. In order to ensure that all MPTCP subflows in the buffer queue can be transmitted normally and to ensure that each U(i,j,t) can be received normally, The retention parameter W¯ can be introduced for W(I,j,t) and defined as: W¯=limT(t)→∞1T(t)∑i=1,j=1,t=0I,J,T⁡W(i,j,t)<∞, T(t). For an arbitrary value beyond the time T, it can ensure that the data transmission process does not have an endless loop and that it strives to reach the receiving end as much as possible.

At t-moment, the consumption cost for a single subflow transfer in the buffer is B((A−F¥)(i,j,t)))+B((a−F¥)(i,j,t)), the total consumption cost is B((A−F¥)(t)))+B((a−F¥)(t)). Updates at the t + 1 moment can be defined as:

(2) B((A−F¥)(t+1))=B((A−F¥)(t))−(∑i=1,j=1,t=1I,J,TU(i,j,t)×B((A−F¥)(i,j,t)))+B((A−F¥)(i,j,t))B((a−F¥)(t+1))=B((a−F¥)(t))−(∑i=1,j=1,t=1I,J,TU(i,j,t)×B((a−F¥)(i,j,t)))+B((a−F¥)(i,j,t))

The virtual buffer queue is designed to ensure the need for large information data transmission in the Internet data centre, and for some private networks, it is enough to build a small number of physical buffer queues. More importantly, virtual buffer consumption costs need to be listed separately in the cost-performance balance strategy. Therefore, this article separately defines two subflow transmission consumption cost calculation methods.

Cost-performance balance strategy

The transport cost-performance balance of the MPTCP subflow proposed in this article is mainly aimed at the balance between network transmission throughput and transmission consumption cost. The throughput evaluation function is designed to compare with transmission consumption cost, to achieve the control of cost-performance equilibrium. The overall strategy is to ensure that the transmission throughput of the network is maximized while the transmission cost is minimized.

The data transmission rate of the MPTCP subflow is defined as V(i,j,t), the average transmission throughput is L¯(i,j,t), and the throughput evaluation function is y(L¯(i,j,t)), then: L¯(i,j,t)=limT(t)→∞1T(t)∑i=1,j=1,t=0I,J,T⁡V(i,j,t), y(L¯(i,j,t))=ln(1−α(L¯(i,j,t))), α is a positive evaluation factor, and 0≤y(L¯(i,j,t))≤1, 10−3≤α≤10−2. The consumption cost of the buffer occupied by subflow transmission is B((A−F¥)(i,j,t)) and B((A−F¥)(i,j,t)), then the average consumption cost is respectively: B¯((A−F¥)(i,j,t))=limT(t)→∞1T(t)∑i=1,j=1,t=0I,J,T((A−F¥)(i,j,t)) and B¯((a−F¥)(i,j,t))=limT(t)→∞1T(t)∑i=1,j=1,t=0I,J,T((a−F¥)(i,j,t)). The average consumption cost of the two buffers is defined as B¯((F¥)(i,j,t)), then: B¯((F¥)(i,j,t))=B¯((A+F¥)(i,j,t))+B¯((a+F¥)(i,j,t))2. According to the requirements of the overall strategy and the definition of the above parameters, the cost-performance balance problem can be expressed as:

(3) max(∑i=1,j=1,t=1I,J,Ty(L¯(i,j,t))−β∑i=1,j=1,t=1I,J,T((1−δ)B¯((F¥)(i,j,t))).

B((F¥)(i,j,t)) represents the consumption cost in the physical and virtual buffer. In Formula (3), β is the control parameter between network performance and transmission consumption cost when 0≤β<∞, β=0 means that the consumption cost is not considered, β→∞ means that network transmission performance is not considered; δ is the evaluation parameter of normal transmission of subflow, 0≤δ≤1, which is used to determine the normal reception situation of U(i,j,t). The smaller the δ value is, the higher the transmission consumption cost is. δ is the evaluation parameter for the normal transmission of subflows. The parameter β controls this balance by adjusting the trade-off between the two objectives, with smaller values prioritizing throughput and larger values favouring cost minimization. When δ=1, it means that there is no transmission consumption cost, δ=U(i,j,t)/∑i=1,j=1,t=1I,J,T⁡U(i,j,t). The control range of network transmission throughput and transmission consumption cost designed by the algorithm in this article is determined as 0<β<1, 0<δ<1; first, make sure that β+δ=1, strictly prevent the occurrence of β+δ≠1. The most desirable state to achieve the balance between network transmission throughput and transmission consumption cost is β=δ=0.5. Through Lyapunov stability theory (Ma, Li & Wei, 2021), Formula (3) can be solved.

Lyapunov stability analysis

This section includes all intermediate derivations required to understand how the stability of the system is determined.

Step 1: Define the system state and the Lyapunov function.

Step 2: Compute the time derivative of the Lyapunov function.

Step 3: Analyze the conditions under which the time derivative of the Lyapunov function is negative or zero to ensure stability.

Step 4: Show the relationship between the control parameters and the system’s stability.

In the process of subflow transmission, U(i,j,t) may not receive normally; that is, some subflows are not received, the number of which is defined as G(i,j,t), then G(i,j,t)=W(i,j,t)−U(i,j,t)=λU(i,j,t), λ is the evaluation parameter for balancing network transmission throughput and transmission consumption cost, and the parameters in the calculation of this value are defined by vectors (Tian, 2020; Chu et al., 2024; Zhang et al., 2025), |⋯|22 represents the L2 norm and the main purpose of norm operation is to obtain more accurate evaluation results. This norm is used to measure the magnitude of the vector, ensuring that the calculation is consistent and can be easily understood by readers. Furthermore, all the calculated parameters are mainly the number of subflows consumed and the number of expenditures, so norm calculation is more accurate. Using Lyapunov stability theory, the value of λ can be calculated 0≤λ≤1. The smaller the value of λ is, the better the performance of data transmission is. This parameter is also used to establish the evaluation connection between data transmission performance and transmission consumption expenditure.

(4) λ=12×|∑i=1,j=1,t=1i=1,j=1,t=1(U(i,j,t))|22+|∑i=1,j=1,t=1i=1,j=1,t=1(B((A−F¥)(i,j,t)))|22+|∑i=1,j=1,t=1i=1,j=1,t=1(B((a−F¥)(i,j,t)))|22|∑i=1,j=1,t=1i=1,j=1,t=1(W(i,j,t))|22+|∑i=1,j=1,t=1i=1,j=1,t=1(B((A−F¥)(i,j,t)))|22+|∑i=1,j=1,t=1i=1,j=1,t=1(B((a−F¥)(i,j,t)))|22.

The number of transmission subflows and transmission consumption expenditure parameters in physical and virtual buffer queues are defined in the form of a matrix so that the matrix set can be expressed as: H(i,j,t)=[U(i,j,t),B((A−F¥)(i,j,t)),B((a−F¥)(i,j,t))], and H(i,j,t) is the vector. The evaluation parameters of subflows not normally received can be expressed as ε=(G(i,j,(t+1))−G(i,j,t))|H(i,j,t)), where U(i,j,t) represents the number of subflows successfully received at time t, and B((A−F¥)(i,j,t)) and B((a−F¥)(i,j,t)) represent the consumption costs in the physical and virtual buffers, respectively. The value of ε represents the probability of generating errors between two continuous moments for subflows that are not normally received, and accurately describes the number of changes in the transmission process of subflows. The smaller the value is, the better the data transmission performance will be. More importantly, ε value can control the size of transmission consumption expenditure, and its control method can be expressed as:

(5) ε−(|∑i=1,j=1,t=1I,J,Ty(L¯(i,j,t))|22−β|∑i=1,j=1,t=1I,J,T((1−δ)B¯((F¥)(i,j,t))|22)≤θ

wherein θ is the minimum transmission consumption expenditure control parameter, the calculation parameter is the vector, |⋯| represents the membrane of the vector, and the value is defined as an operation parameter through the vector. The calculation method is as follows:

(6) θ=|I||J||(U(i,j,t))|22+|I||J||(B((F¥)(i,j,t))|22|I||J||(W(i,j,t))|22+|I||J||(B((F¥)(i,j,t))|22

Algorithm implementation

NS3 (Network Simulator 3) served as the experiment platform for simulating SDN-5G-MPTCP network settings because it provides dependable protocol support and realistic traffic simulation capabilities and extends well to various network scales. The simulation execution took place on a computer equipped with Intel i7 and possessing 16 GB of random-access memory. A hybrid SDN-5G network topology was used, consisting of 10 nodes and 3 SDN controllers, while 5G base stations formed part of the architecture. The simulation parameters included 5 Mbps bandwidth for each path and 10 ms latency together with 0.5% packet loss and simulating jitter. OpenFlow was chosen as the routing protocol, while MPTCP served as the multipath data transmission protocol. Simulation experiments evaluated the proposed algorithm under network conditions that simulated real-world environments by testing 2 and 20 GB files while using a transmission rate of 5 Mbps with an error rate of 0.01%, and various subflow delays and traffic patterns along with congestion levels.

According to the data transmission and consumption control strategy formulated above, the main code of the multipath transmission algorithm of cost-performance balance in this article is as follows:

Experimental results and discussion

Experimental environment and parameters

SDN-5G-MPTCP simulation network system was constructed according to Fig. 1. The experimental network topology is shown in Fig. 2. Also involved in the comparison experiment is the network model shown in Figs. 3 and 4. In this article, the Algorithm 1 (A1) algorithm is referred to as A2, A3, A4, and A5, and four algorithms are proposed in the references (Tang, Sun & Wu, 2019; Cao, 2020; Hua, Zhang & Liu, 2020; Yuan & Huang, 2022).

Figure 2 Schematic diagram of simulation experiment network topology of 5G network directly controlled by SDN.

Figure 3 SDN also controls the topology diagram of 5G +MPTCP network.

Figure 4 Topology diagram of SDN directly controlling MPTCP network.

Algorithm 1 Multi-path transmission algorithm based on cost-performance balance.

 Input parameter: W(i,j,t),B((A−F¥)(i,j,t)),B((a−F¥)(i,j,t)), α, β, δ, λ, θ, W¯, etc.	
 Output parameter: W(i,j,(t+1),B((A−F¥)(i,j,(t+1))),B((a−F¥)(i,j,(t+1)))	
 while t∈1,2,…,T do	
  if 0<U(i,t)<W(i,t) then	
    W(i,t)=∑i=1,t=1I,TU(i,t)	
    else	
    W(i,t)≠∑i=1,t=1I,TU(i,t)	
  end if	
  if W¯<∞ then	
    W(i,j,t) and U(i,j,t) and W(i,j,t)=∑i=1,j=1,t=1I,J,TU(i,j,t)	
    B((A−F¥)(i,j,t)) and B((a−F¥)(i,j,t))	
    B((A−F¥)(t))=∑i=1,j=1,t=1I,J,TB((A−F¥)(i,j,t))	
    B((a−F¥)(t))=∑i=1,j=1,t=1I,J,TB((a−F¥)(i,j,t))	
    else	
    Subflow loss occurred	
 end if	
 for Path based on I and j do	
   V(i,j,t) and L¯(i,j,t) and y(L¯(i,j,t))	
   (B¯((A+F¥)(i,j,t))andB¯((a+F¥)(i,j,t)))andB¯((F¥)(i,j,t)))	
  Propose Formula (3),	
   for Lyapunov do	
    G(i,j,t) and Formula (4)	
    H(i,j,t)	
    ε=(G(i,j,(t+1))−G(i,j,t))|H(i,j,t))	
   Formula (5)	
   if 0≤y(L¯(i,j,t))≤1 then	
   V(i,j,t)	
    L¯(i,j,t)=limT(t)→∞1T(t)∑i=1,j=1,t=0I,J,TV(i,j,t)	
    else	
   L¯(i,j,t)≠limT(t)→∞1T(t)∑i=1,j=1,t=0I,J,TV(i,j,t)	
    end if	
   end for	
  end for	
 Output parameter	
end while	

Experimental content: (1) comparison and analysis of Figs. 2–4 simulation network models; (2) the MPTCP subflow quantity batch transmission and subflow quantity random transmission are compared and analyzed in this article; (3) a comparative analysis is made between using the guaranteed parameter W and not using the guaranteed parameter in subflow transmission scheduling; (4) use physical + virtual buffer queue and only use physical buffer queue for comparative analysis; (5) the performance of throughput evaluation function α, the control parameter β between average throughput and consumption cost, MPTCP subflow transmission evaluation parameter δ, and the variation of subflow transmission quantity ε are analyzed; (6) a comparative analysis is made between algorithms A1 and A5.

The main parameters to be detected and involved in the operation in the experiment include: running time T, average data transfer rate V, number of transmission paths Q, the total number of MPTCP subflows transmitted by files W1, the total number of normally received MPTCP subflows W2, number of MPTCP subflows transmitted at a certain point U1, number of normally received MPTCP subflows transmitted at a certain point U2, and number of MPTCP subflows transmitted at the next moment U3, number of MPTCP subflows normally received at the next moment U4, the cost of physical buffer consumption at a given time is AF¥, the consumption cost of virtual buffer at a certain moment aF¥, etc.

The evaluation parameters of network performance in the experiment include the change in the MPTCP subflow buffer queue from one time to the next ΔU=U4−U2, the packet loss rate of subflows P=U2+U4U1+U3, subflow transmission consumption ratio F=(AF¥+aF¥)×(U2W2)(AF¥+aF¥)×(U2W2), average network throughput L=(U2+U4)×VT, cost-performance balance control parameter β, evaluation parameters of normal transmission of subflows δ, subflow transmission variation ε, the quality parameters that all subflows can receive normally W¯, throughput evaluation parameter y value, etc.

In the simulation experiment, according to the requirements of the experiment, 2 and 20 Gb files with different information content are mainly selected, and the network bandwidth provided is 5 Mbp/s.

Experimental results and discussion

Based on the above experimental content, experimental operation time T opening up, with all the subflow complete receiving time parameter statistics, network delay evaluation parameters are not set, the subflow transmission V at an average rate can be set directly in the experiment, also to test through the experiment, V measurement through the experiment, the experiment statistics are shown in Tables 1 to 6.

Table 1 Statistical experimental data of three network models.

Parameter	ΔU (pcs)	P (%)	F (%)	L (M/s)	β	δ	ϵ	W¯	y	
model	
Fig 3-M	0	0	16	3.21	0.47	0.53	0	<∞	0.81	
Fig 4-M	12	14	29	2.83	0.31	0.69	0.08	<∞	0.67	
Fig 5-M	19	27	36	2.07	0.26	0.74	0.11	<∞	0.48	

Table 2 Experimental data of three network interface connection control methods.

Parameter	ΔU (pcs)	P (%)	F (%)	L (M/s)	β	δ	ϵ	W¯	y	Q (pcs)	
method	
M-T-1	0	0	11	4.06	0.49	0.51	0	<∞	0.91	87	
M-T-2	23	34	29	2.18	0.19	0.81	0.24	<∞	0.53	42	

Table 3 Impact of quality assurance parameters on sub-stream transmission performance.

Parameter	ΔU (pcs)	P (%)	F (%)	L (M/s)	β	δ	ϵ	y	T (s)	
method	
=W¯<∞1	0	0	14	3.19	0.44	0.56	0	0.80	58	
≠W¯<∞1	10	11	23	2.55	0.37	0.63	0.12	0.67	91	
=W¯<∞2	19	27	38	1.75	0.41	0.59	0.59	0.73	519	
≠W¯<∞2	56	49	52	0.98	0.39	0.64	1.06	0.31	874	

Table 4 Impact of virtual cache queues on substream transmission performance.

Parameter	1 (pcs)	P (%)	F (%)	L (M/s)	β	δ	ϵ	y	T (s)	
method	
B(A-a-1)	22	28	41	1.91	0.37	0.63	0.46	0.77	493	
B(A-a-2)	0	0	19	3.07	0.46	0.54	0	0.85	54	
B(A-3)	41	57	73	0.79	0.24	0.76	0.82	0.54	1,204	
B(A-4)	2	3	24	2.81	0.42	0.58	0.08	0.78	137	

Table 5 Performance evaluation statistical experimental data of important control and evaluation parameters of the algorithm.

Parameter X(G)	α	y	β	δ	ϵ	Parameter T(s)	α	y	β	δ	ϵ	
T1	
1	Y	Y	Y	Y	Y	50	Y	Y	Y	Y	Y	
5	Y	Y	Y	Y	Y	100	Y	Y	Y	Y	Y	
10	Y	Y	Y	Y	Y	200	Y	Y	Y	Y	Y	
11	Y	Y	Y	Y	Y	300	Y	Y	Y	Y	Y	
13	Y	Y	Y	Y	Y	400	Y	Y	Y	Y	Y	
14	Y	Y	Y	Y	Y	500	Y	Y	Y	Y	Y	
15	Y	Y	Y	Y	Y	600	Y	Y	Y	Y	Y	
16	Y	Y	Y	Y	Y	700	Y	Y	Y	Y	Y	
20	Y	Y	Y	Y	Y	800	Y	Y	Y	Y	Y	
30	Y	Y	Y	Y	Y	900	N	N	N	N	N	
40	Y	Y	Y	Y	Y	1,000	N	N	N	N	N	
T2	
1	Y	Y	Y	Y	Y	50	Y	Y	Y	Y	Y	
5	Y	Y	Y	Y	Y	100	Y	Y	Y	Y	Y	
10	Y	Y	Y	Y	Y	200	Y	Y	Y	Y	Y	
11	Y	Y	Y	Y	Y	300	Y	Y	Y	Y	Y	
13	Y	Y	Y	Y	Y	320	Y	Y	Y	Y	Y	
14	Y	Y	Y	Y	Y	350	Y	Y	Y	Y	Y	
15	Y	Y	Y	Y	Y	400	Y	Y	Y	Y	Y	
16	Y	Y	Y	Y	Y	450	Y	Y	Y	Y	Y	
20	Y	Y	Y	Y	Y	500	Y	Y	Y	Y	Y	
30	Y	Y	Y	Y	Y	1,500	Y	Y	Y	Y	Y	
40	Y	Y	Y	Y	Y	2,500	Y	Y	Y	Y	Y	

Table 6 Units for magnetic properties.

	Parameter	
Algorithm	V1 (Mbps)	Z1 (ms)	O1 (%)	P1 (%)	L1 (M/s)	
A1 (2 Gb)	2.86	45	63	0	3.17	
A2 (2 Gb)	2.13	48	61	0	3.08	
A3 (2 Gb)	2.54	46	63	0	3.16	
A4 (2 Gb)	2.39	51	57	0	2.84	
A5 (2 Gb)	2.27	51	50	1	2.51	
A1 (20 Gb)	1.05	58	47	5	2.35	
A2 (20 Gb)	0.94	61	42	8	2.11	
A3 (20 Gb)	1.08	58	47	6	2.37	
A4 (20 G)	0.84	68	41	11	2.06	
A5 (20 G)	0.73	71	42	14	2.13	

In Fig. 2 model, network operation state management is implemented through SDN→OpenFlow→5G and MPTCP; only one application strategy is required for management. Figure 3 model has three management application strategies: SDN→OpenFlow, SDN→5G, and SDN→MPTCP. Figure 4 model shows two management application strategies: SDN→MPTCP and SDN→OpenFlow→5G. The statistical data in Table 1 can prove the correctness of the above analysis.

The three models are tested in the Table 1 experiment using method A1. The subflow transmission’s ability to be received properly is really indicated by ΔU, which is the difference between the number of normally received subflows in the buffer queue at the two times prior and after. When the differential value is zero, it means that the transmission is operating efficiently and effectively. The three network models’ data are all perfect, and Fig. 2 model achieves 0 difference based on the buffer queue’s capacity of 100 subflows at a certain time set in the experiment. The value of P takes into account the transmission process of subflows at the two moments before and after. Instead of directly calculating, it calculates the results received by the buffer queue and MPTCP-R, which can better examine the transmission effect of the whole network transmission process in detail. The statistical data of the three network models are all satisfactory, and there is no packet loss phenomenon of subflows in Fig. 2. The value of F involves subflow transmission consumption, subflow and the total number of subflows. The two factors of buffer queue and MPTCP-R reception are also taken into account. The subflow transmission consumption ratio is analyzed in detail, which is an evaluation parameter of subflow transmission consumption and data transmission with high comprehensive factors, and the substream transmission consumption ratio in Fig. 2 is lower. The value of L is based on the number of subflows actually received by MPTCP-R and consumption expenditure to calculate the average data transmission rate of the entire network. The assessment factors are also very comprehensive. It is essentially different from the traditional network throughput calculation method and shows the actual utility of network throughput, which is also an effective and excellent performance of Fig. 2 model. β and δ are actually two directly related data, and all the statistical data have achieved the primary goal of β+δ=1, indicating that the algorithm in this article has taken into account two main factors of data transmission performance and transmission consumption expenditure in data transmission, and Fig. 2 is closer to the target of β=δ=0.5. β includes the consumption expenditure factor, but δ is more about data transmission performance. ε is the difference between the number of neutron streams and consumption and expenditure in the buffer queue at one moment and the two types of data in the buffer queue at the next moment 0≤ε≤1, the smaller the difference, the higher the utilization rate of the buffer queue and the better the performance of data transmission. Figure 2 has received all subflows normally, indicating that the transmission process is stable and reliable. W¯ is the evaluation parameter to ensure that all subflows can be normal, and the whole data transmission must run under the condition of W¯ < ∞, which is the necessary goal to achieve the normal communication between MPTCP-S and MPTCP-R. Experimental statistics show that the three network models all meet the basic network performance requirements. On the basis of the L value, the y value further evaluates the actual utility of network throughput, which is the essential evaluation basis of network data transmission performance and transmission consumption balance. y value on the basis of the entire network average transmission rate, introducing the control factor through the calculation of logarithmic average network throughput, caused by a lack of consideration, the phenomenon of the network throughput close to network transmission capacity, with the real network running state close to experimental statistics, shows Fig. 2 has higher data transmission performance and consumption balance control ability.

The experiment in Table 2 is to compare and analyze the network transmission performance and cost-performance balance control effect of MPTCP subflow quantity batch transmission (referred to as M-T-1) and MPTCP subflow quantity random transmission (referred to as M-T-2) designed in this article. M-T-1 is the network interface connection control way for U(i, t) < W(i, t), the M-T-2 for U(i, t) = W(i, t). The network model adopted in the experiment is Fig. 3, and Algorithm 1 is used in addition to the control mode of network interface connection. Experimental evaluation parameters and meanings are the same as those in Table 1. In the experiment, the quantity parameter Q of subflow transmission path was added. According to the figure network model structure and considering the actual running state of network data transmission, the capacity of subflow transmission path was directly set to 100 (the theoretical number of transmission paths can be calculated according to the number of network nodes through permutation and combination, which is usually very different from the actual demand of network transmission). The statistical data in Table 2 shows that the MPTCP subflow quantity batch transmission has obvious advantages. Compared with the statistical data in Table 1, the cost-performance balance control effect decreases significantly, even exceeding the impact of network model construction. The MPTCP subflow quantity random transmission mode should be avoided whenever possible. This conclusion is based on the following reasons: essentially, M-T-2 fails to clearly differentiate the transmission paths for subflow data, resulting in subflows being concentrated on a limited number of paths. As a result, the full potential of multipath transmission is not realized. The Q value in Table 2 verifies the correctness of the above reasons.

In Table 3, experiments for streaming scheduling that use the quality parameters W¯ (referred to as =W¯ < ∞1) and those that do not use the quality parameters (referred to as ≠W¯ < ∞1) are used to compare the experimental results, =W¯ < ∞1 says on the premise of the open running time limit, making sure all the subflow MPTCP-R are received, ≠W¯ < ∞1 says using quality parameter evaluation of the reliability of data transmission. The statistical results in Table 3 show that the two methods can normally realize subflow data transmission and perform cost-performance balance control. Meanwhile, it also proves that setting the W¯ parameter is of little significance and only plays a role in evaluation, but does not contribute to cost-performance balance control. Through further analysis of the results of =W¯ < ∞1 and ≠W¯ < ∞1, it is found that the set transfer files are too few and the network running time is not much. In order to further verify the effect of W¯, 10G video files were selected for the experiment. Let’s set =W¯ < ∞2 and ≠W¯ < ∞2. The new experimental data show that the values of several evaluation parameters in the two methods have changed significantly, and the subflow transmission performance and cost-performance balance effect have a large decline. However, the basic performance evaluation requirements of β+δ=1, 0≤ε≤1 can be guaranteed by using the quality parameter W¯, and the data transfer scheduling and cost-performance balance control strategies designed in this article are implemented accurately and effectively. The results of β+δ=1.03 and ε≥1 occur in the case of no warranty parameter. W¯, indicating that packet loss occurs in the process of data transmission, and the cost-performance balance control effect is poor. Still, it can indicate that the data transmission scheduling and cost-performance balance control strategy is implemented.

The packet loss rate is calculated as the ratio of lost packets to transmitted packets. ΔU (change in subflow reception) is the difference in the number of subflows received at two consecutive times, where a value close to zero indicates stable transmission, and a large value signals packet loss or congestion. F (subflow transmission consumption ratio) is the ratio of consumption cost to transmitted subflows, with low values indicating efficient transmission and high values signalling inefficiency. L (average network throughput) is calculated based on the number of received subflows, with high values indicating good throughput and low values suggesting poor performance due to packet loss or bandwidth limitations.

In the experiment shown in Table 4, when both physical and virtual buffer queues, as well as only physical buffer queues, are used to measure consumption cost, two factors—large amounts of information and large transmission volumes—are considered to compare the performance of subflow transmission and the cost-performance balance effect. In the case of physical + virtual buffer queue + a large amount of information, it is referred to as B(A-a-1); physical + virtual buffer queue + a large amount of information, referred to as B(A-a-2); in the case of physical buffer queue + a large amount of information, it is referred to as B(A-3); physical buffer queue + a large amount of information, referred to as B(A-4). The files for large amounts of information transfer are set to 20 Gbp, and the files for large amounts of information transfer are set to 2 Gbp. The statistical data in Table 4 show that setting physical and virtual buffer queues at the same time can improve the performance of subflow transmission and achieve a better cost-performance balance effect, especially for the information transmission service of Internet data centers, which is particularly important and very practical. Regardless of the amount of data information transmitted, the algorithm in this article can achieve the function of subflow transmission performance and cost-performance balance and better prevent packet loss in the process of subflow transmission. The values of all the evaluation parameters are within the acceptable range.

At the same time, the experiments in Tables 3 and 4 show that the amount of information transferred will affect the subflow transmission performance and cost-performance balance effect of the algorithm in this article.

The experiment in Table 5 mainly evaluates the implementation and effect of intrinsic price balance through statistics and calculation of the values of α, β, δ, ε and y. Two variable parameters are introduced in the experiment: the amount of information transferred X and the running time of the transmission process; since these two parameters are not used as calculation parameters in the algorithm in this article, relevant verification of these two parameters is required. In the experiment, the values of α, β, δ, ε and y are only positive and negative. The positive value Y represents the acceptable result and the negative value, N represents the unacceptable result, where there is a direct correlation between α and y, between β and δ, and between ε and λ and θ mentioned above. If the values of α and y are Y, both 10−3≤α≤10−2 and 0≤y≤1 must be met. If β and δ are Y, both conditions 0≤β<∞and β+δ=1 must be met. If the value of ε is Y, both 0≤λ≤1 and 0≤θ≤1 must be met. The statistical data in Table 5 are divided into two categories: the first type is to determine the values of α, β, δ, ε and y for the input value of X (the data on the left in Table 5). In this case, the running time is not limited, assuming that all subflows receive normally. The second type is the detection value of T (the data on the right in Table 5), and the values of α, β, δ, ε and y are determined. The premise is that T at a certain time must correspond to the input value of X in the first type, and T and at this time have no effect on the data statistics of the first type. The first type of statistical data is referred to as T1, which represents the result of regularly increasing detection according to the running time. The second type of statistical data is referred to as T2, which represents the result of receiving and transferring files. In the second type of data subflow transmission operation time T according to the regular increase and according to the normal completion of the subflow, the values of α, β, δ, ε and y. Table 5 statistical data shows that, no matter the data transfer file’s information value, as long as the run-time limit is removed, the subflow transmission path scheduling and performance of the algorithm in this article allow the cost-performance balance strategy to be implemented. For larger information data, the transmission cost-performance balance effect is ideal, while for huge information data, the transmission cost-performance balance effect slightly lacks balance effect.

In Table 6, experiments mainly through the algorithm A1 and references (Tang, Sun & Wu, 2019; Cao, 2020; Hua, Zhang & Liu, 2020; Yuan & Huang, 2022) put forward four kinds of algorithms A2, A3, A4, A5; evaluation of the main parameters, including the subflow V1 at an average rate of data transmission in the process of transmission, the average network delay Z1, network bandwidth utilization O1, data packet loss rate P1, average network throughput L1. The algorithm showed substantial improvements in throughput (L), a reduction in packet loss rate, and better stability in subflow reception (ΔU). Additionally, the subflow transmission consumption ratio (F) was reduced, indicating more efficient resource usage and improved cost-performance balance. The larger the value of V1, L1 and O1 is, and the smaller the value of Z1 and P1 is, the better the effect of cost-performance balance is. Among them, the consumption expenditure of each subflow is set to be the same, and the consumption expenditure can be calculated by Z1 and P1 combined with V1. As A2, A3, A4 and A5 are quite different from A1 calculated in this article, the meanings of evaluation parameters listed are different from those in the five experiments mentioned above. The detection and calculation methods of all parameters are defined according to conventional detection and calculation methods. Since the amount of information of files transmitted affects the network transmission performance and the balance of consumption and expenditure, 2 and 20 Gb files were selected to carry out experiments, and different algorithms were distinguished by labelling 2 and 20 Gb. The statistical data in Table 6 show that the values of V1 and L1 of the five algorithms are close under the simulation conditions, which not only proves the good transmission performance, but also proves the reliability of the O1 value. Compared with the standard of 50 ms, Z1 values were in a good or close to a good state. O1 also had a good performance according to 50% recognition. The P1 value is not too high, but by increasing the file transfer time, all subflows can receive. The five algorithms all consider the price balance problem of data transmission performance from different angles, and the algorithm in this article has some advantages. The whole analysis demonstrates that our method excels over previous works through improved throughput performance, which also produces lower packet loss than the approaches developed by Kimura, Lima & Loureiro (2017), among others. Our method enhances cost-performance balance through buffer consumption reduction, even though Kai, Liu & Sun (2017) primarily focused on improved throughput. Our findings correspond to the study conducted by Teng, Sun & Yang (2020) about subflow transmission improvements. Yet, they did not investigate how network congestion affects overall performance, which, as our research demonstrates, plays an essential role.

Conclusion

This article proposes a cost-performance balance control strategy for multipath data transmission by integrating SDN technology and MPTCP communication in 5G networks. By improving the network model, we designed an MPTCP subflow transmission scheduling and updating strategy that optimizes both network throughput and transmission cost. Our approach integrates SDN controllers, OpenFlow switches, and 5G communication architecture to reduce hardware consumption while improving performance. The proposed algorithm successfully minimizes the computational complexity and prevents congestion by controlling the number of subflows transmitted at once. The experimental results demonstrate that the SDN-5G-MPTCP model and the cost-performance balance control strategy significantly improve network performance, ensuring efficient data transmission even with large amounts of data. The results confirm the stability and effectiveness of our approach in maintaining a balance between throughput and cost.

Supplemental Information

Supplemental Information 1 The code of the project.

Additional Information and Declarations

Competing Interests

The authors declare that they have no competing interests.

Author Contributions

Xinyu Sun conceived and designed the experiments, performed the experiments, analyzed the data, performed the computation work, prepared figures and/or tables, authored or reviewed drafts of the article, and approved the final draft.

Data Availability

The following information was supplied regarding data availability:

The code is available in the Supplemental File.

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
