# Peer review of "Multipath subflow transmission scheduling optimization algorithm based on cost-performance balance"

_PeerJ Computer Science, doi:10.7717/peerj-cs.2838_

## Round 0.1 · original submission · Major Revisions

Please see both reviewers' comments. The feedback emphasizes improving clarity and consistency in the paper. Key suggestions include simplifying the abstract, correcting grammatical errors, standardizing notations, and reformatting equations for better readability. Detailed explanations for complex concepts, such as Lyapunov stability theory and cost-performance balance, are recommended. The experimental section should provide more details on the simulation setup, and tables and figures should have clearer captions. Diagrams and step-by-step breakdowns can help illustrate complex ideas, and performance metrics should be clearly defined. Proper formatting of pseudo-code and integrating references more effectively are also advised.

Reviewer 1 ·

Basic reporting

1. The abstract contains long, complex sentences. Consider breaking them into shorter sentences to clarify the proposed algorithm’s key contributions (e.g., “Design a unified communication interface…” can be revised for clarity).
2. Several sentences have grammatical errors and awkward phrasing (e.g., “sub-s ε tream update” should be “subflow update”). A thorough proofreading for grammar and clarity is recommended.

Experimental design

3. Notations like B((A-F¥)(i,j,t)) appear inconsistent. Standardize symbols throughout the paper to avoid reader confusion. Define each symbol clearly when it first appears.
4. Equations such as (1) and (3) are densely packed. Reformat these equations with clear spacing and numbering. Consider writing intermediate steps or explanations to enhance readability.
5. Several parameters (α, β, δ, ε, λ, etc.) are introduced without a consolidated explanation. A notation table summarizing definitions, ranges, and roles of each parameter would be very helpful.
6. The paper uses vectors and matrices (e.g., H(i,j,t)) in the analysis. Ensure that the operations (e.g., L_2 norm, membrane notation) are clearly defined and that the notation is consistent throughout the manuscript.

Validity of the findings

7. The application of Lyapunov stability theory is mentioned to solve equation (3). However, the paper does not show intermediate derivations. Consider adding an appendix or a detailed section outlining the steps in the stability analysis.
8. The cost-performance balance problem (equation (3)) is quite complex. Consider breaking it down into simpler sub-problems or providing additional explanation to make the underlying optimization goal clearer.

Reviewer 2 ·

Basic reporting

1. The experimental section describes simulation results but does not detail the simulation setup (tools used, network conditions, etc.). Providing more background on the simulation environment will aid reproducibility.
2. The tables (TABLE I–VI) contain valuable data, but their captions and labeling could be more descriptive. Ensure each table clearly explains the experimental parameters and performance metrics.
3. The paper’s technical depth is impressive, yet the overall structure could benefit from a more reader-friendly flow. Consider reorganizing sections (e.g., by merging similar topics or highlighting key results in dedicated subsections) to guide the reader more effectively through the complex material.
4. The paper mentions “2 GBP” and “20 GBP” files, which may be confused with currency units. Clarify that “GBP” stands for gigabytes or adjust the notation accordingly to avoid ambiguity.

Experimental design

5. The description of physical and virtual buffer queues and their cost functions is dense. Consider using diagrams or step-by-step breakdowns to illustrate how these costs are computed and compared.
6. In the discussion on subflow scheduling, the differences between batch transmission (M-T-1) and random transmission (M-T-2) are noted. Enhance this section with a more detailed explanation of why one method outperforms the other, possibly with a graphical comparison.
7. The definition of packet loss rate and related performance metrics (e.g., ΔU, F, L) should be clarified. Provide formulas and intuitive explanations of what constitutes “normal” versus “abnormal” transmission.
8. While experimental data is provided in multiple tables, the discussion is somewhat repetitive. Summarize the key insights succinctly and consider highlighting which performance improvements are most significant.

Validity of the findings

9. Algorithm1 is presented in a pseudo-code format with some inconsistencies in indentation and structure. A more structured pseudo-code with proper indentation and clear comments for each block would improve clarity.
10. Figures (e.g., Figures 1, 2-M, 3-M, 4-M) are referenced with minimal descriptions. Enhance captions to fully explain the contents and relevance of each figure to the proposed method.
11. The paper contains several grammatical errors, including awkward phrasing and punctuation issues. I recommend revising sentences for clarity and ensuring subject-verb agreement throughout.
12. The references are extensive, but the discussion sometimes lacks clear integration with related work. Consider drawing clearer connections between your contributions and those of the cited literature.
13. The conclusion section recaps the technical approach but could be strengthened by summarizing key findings, discussing the limitations, and suggesting directions for future work.

---

## Round 0.2 · accepted · Accept

Both reviewers have confirmed that their comments have been addressed.

Reviewer 1 ·

Basic reporting

no comment

Experimental design

no comment

Validity of the findings

no comment

Reviewer 2 ·

Basic reporting

The improved version of article seems to be well versed and in acceptable position right now.

Experimental design

While overall paper is good, the experimental design gives a quite good understanding on methods and principals proposed in this article.

Validity of the findings

Results are effectively understandable as compare to the claim